# Young Carers in Early Childhood—How Are Young Carers Represented in Broader Literature and What Factors Influence Dominant Representations of Young Carers in Early Childhood in the UK?

**DOI:** 10.3390/healthcare13030280

**Published:** 2025-01-30

**Authors:** Carly Ellicott, Sayyeda Ume Rubab, Andy McGowan, Beth Neale, Ali Bidaran, Felicity Dewsbery, Alyson Norman, Helen Lloyd

**Affiliations:** 1Faculty of Health, School of Psychology, University of Plymouth, Plymouth PL4 8AA, UK; sayyeda.umerubab@plymouth.ac.uk (S.U.R.); ali.bidaran@plymouth.ac.uk (A.B.); alyson.norman@plymouth.ac.uk (A.N.); helen.lloyd-1@plymouth.ac.uk (H.L.); 2Carers Trust, 2-6 Boundary Row, London SE1 8HP, UK; amcgowan@carers.org (A.M.); bneale@carers.org (B.N.); 3Pen Green Centre for Children and Their Families, Corby NN171BJ, UK; felicity.dewsbery@northnorthants.gov.uk

**Keywords:** young carers, early childhood, early years, prevention

## Abstract

Contextualization of young carers in early childhood is complex. Biopsychosocial impacts of young caregiving are receiving growing interest, yet the voice and experience of society’s youngest carers (0–5 years) remain absent from the literature. This scoping review explores representation of young carers in their early years (0–5 years), presented as part of a broader program of PhD research undertaken by the lead author. Aiming to influence systematic change in the way young carers are perceived and supported in society, further research is recommended. This will serve to better inform whole family support strategies in the context of young carer policy and practice. This review has been undertaken following JBI guidance for scoping reviews. A comprehensive literature search included publications dated 2014–2024 in Scopus, PubMed, PsycINFO, Eric, Web of Science, and Carers Trust and NSPCC websites. The overall search yielded no studies which met the inclusion criteria. Results were discussed with knowledge users, and content experts with lived experience of caring in early childhood, at all stages of the review process. The absence of relevant research highlights a significant gap in knowledge regarding the way in which the lived experiences of young carers in early childhood are represented and understood.

## 1. Introduction

Conservative global estimates anticipate 2–8% of the world’s children and young people have a level of caring responsibility [1]. In context, 3.55 million children living in the UK are currently in their early years (0–5) [2]. Evidence suggests for some young carers that their role is assumed between three and ten years before they are identified [3]. Whilst recognition of young caregiving has undoubtedly grown exponentially in recent decades, it is widely acknowledged that young carers are still not identified at the earliest possible opportunities [4,5]. In the UK, recent household census data identified young carers as young as three years old providing unpaid care in Scotland [6]. Furthermore, prevalence of young carers recognised from the age of four years is evidenced through provision of dedicated young carers services across the UK [7,8]. Evidence suggests a 3% rise in prevalence from 8% to 9.8–11.9% following the COVID-19 pandemic in analysis of three longitudinal data sets of young carers aged 16–18 years old [9]. For many of these young people, it is likely that their caring role began in early childhood.

The contextualization of children as caregivers is a complex matter [10,11]. The psychosocial impacts of young caregiving have received growing interest in recent years with qualitative research suggested to have reached saturation in some respects [12]. Still, the voice of young carers in early childhood (YCEC) is seldomly represented in the broader field of empirical research [13], particularly the voice of society’s youngest carers (0–5 years). Whilst not limited to the context of young carers, voicing experiences of our youngest children is of fundamental importance [14,15], both to inform policy and to foster pathways to achieving national and international aspiration for improved young carer identification and support [16,17].

Definition in context: Global typology of young caregiving is not categorised by a single unified definition. Mutual understanding views a young carer as any child or young person under the age of eighteen providing or intending to provide care to another person with an illness, addiction, disability, or mental health condition.

“Young carers in every country look after someone in their family who has an illness or a disability or other condition. Sometimes they look after the whole family. Young carers are children and young people first and should be free to develop emotionally and physically and to take full advantage of opportunities for educational achievement and life success” [17] (p. 1).

In the United Kingdom (UK) ‘Young Carer’ is used unanimously within legislation in England, Children Act 1989, as modified by the Children and Families Act 2014 [18]; Health and Care Act, 2022 [19], Scotland (The Carers (Scotland) Act 2016) [20], and Wales (The Social Services and Well-being (Wales) Act 2014) [21]. Young carers are recognised in Northern Ireland under a broader definition of children in need, with ‘Young Carer’ used within practice guidance [22,23]. Legislation throughout England, Scotland and Wales embeds the right for all children and young people, regardless of age to an assessment of need. Northern Ireland enacts this right through the Children (Northern Ireland) Order 1995 [24].

Early childhood is globally typified as the period from birth to school entry, which varies across the world [25,26]. In the UK, infancy, early years and early childhood are overlapping constructs with distinct age ranges within the concept of early childhood. In this context, three intersecting, yet distinct, developmental constructs exist: infancy, the period from conception to the age of two years [27], and early years and early childhood, which are used interchangeably to describe young children aged from birth to five years [28,29]. In the context of this review, and with commitment to reduce research waste [12] the focus centers on the representation of YCEC aged from birth to five years [30].

Optimal child development forms the basis on which community adhesion and economic growth are founded to build thriving, sustainable societies [31]. The Sustainable Development Goals (SDG), underpinned by the leave no one behind (LNOB) principle [32], address the crucial factors of poverty, (SDG1), health and wellbeing (SDG3) and education (SDG 4) [33], each having a significant impact upon young carer well-being and life opportunities. Establishing equitable opportunities for YCEC to achieve these goals is of imperative importance, helping to ensure that those furthest behind are receiving support at the earliest opportunities. Importantly, UK-wide investment in the early childhood care and education (ECCE) sector has taken bold steps in recent times to address inequalities. In England, policy seeks to increase access to provision, starting in areas deemed as ‘childcare deserts’. Evidence indicates that 45% of children under the age of five years in England are currently not able to gain access to provision in their local area, and as a result existing childcare funding entitlements are less accessible to lower income families, exasperating availability of support to families in need [34] (Figure 1), in particular for those most vulnerable and needing care.

Commitment to investment in ECCE is unified across each of the four nations (UK). A report recently commissioned by The Royal Foundation of The Prince and Princess of Wales business taskforce for early childhood estimates that £45.5 billion of value would be added each year to the UK economy by investing in early childhood [35] (p. 8). Such commitment is mirrored within the health care system with the Health and Care Act (2022) [19] prioritising young children’s health and wellbeing within Integrated Care Boards (ICBs) across England. The Children and Young People’s Health Policy Influencing Group have designed a visual pathway, which sets out a vision centering children within the core of all health care provision [36] (Figure 2). Its unifying commitment to these promises obligates policy to reflect the vital role the ECCE sector plays within the puzzle of system change concerned with young carer and whole family support. In doing so, professional awareness across the sectors of health and social care and ECCE harnesses the potential to recognise and prevent inappropriate caring responsibility at the earliest opportunity and, in addition, strengthening young children’s transition into primary education, where early childhood development continues to be supported; yet outcomes for young carers are significantly impacted in comparison to their non-caregiving peers [37].

The framing of YCEC in the UK sits within the context of interdisciplinary, whole family support, fundamental to the reduction of negative outcomes associated with the care of another person [38]. Whole family support serves to strengthen the family unit, placing emphasis on growing secure, consistent, supportive relationships [39]. Aligning with principles of early childhood curricula, attachment theory informs relational pedagogy, fostering a sense of belonging and companionship within child, family, and professional triads [40,41]. Developing healthy attachments in early life is therefore central to the moderation and reduction of childhood stress and associated bio-psychosocial outcomes in later life [42,43,44]. The Think Family model helps demonstrate this point, as developed by Yghemonos [45] (p. 55), from the Department for Education’s Think Family Toolkit [46]. Whole family support is essential to mitigating poor outcomes derived from often time-limited and isolated support.

When child caregiver relationships become fractured, a ripple of insecurity, discomfort and compromised feelings of safety extend throughout the ‘nested structures’ which encompass the child’s unique lived experience [47]. In this regard, successful support and intervention through adoption of a ‘virtuous circle model’ takes account of a child’s wider sociocultural ecosystem, as such fostering companionship and helping to reinforce mutually resilient attachments in early childhood and beyond.

O’Dea and Marcelo [48] discuss the experience of caregiver support in the broader context of caregiving amongst adult populations. Importantly, they warn that having support accessible throughout the layered systems of education, health, social care, and wider community contexts does not necessarily mean carers feel supported, concurring with Ellicott et al. [49] that ‘think family’ approaches, whilst advocated within social policy, are yet to be fully adopted consistently into practice [38] (p. 21). To address this, Lindley [50] argues for a ‘National Children’s Service’ which would place whole family systems at the core of all service delivery. Mirrored by the ideals set out by the Carers Trust [8] in its recently updated *No wrong doors for carers* memorandum of understanding, accountability for the identification and support of young carers (and their families), therefore, sits at the door of all health, care, and education professionals, including the ECCE sector, service commissioners and strategic leads. Entwined whole family systems thus serve to mitigate isolation, not only for the child and family, but for organisations working in isolation of one another, creating silos.

Young carers (including children who may become young carers) within the UK have a right to voice their experience and have their needs assessed. These rights must be protected in the context of safeguarding and equitable opportunity for all young carers [51,52,53]. Drawing examples from practice in England, the *Framework for the Assessment of Children in Need and their Families* [54] served as a guiding compass for all assessments under the umbrella of the Children Act 1989 [55] (as modified by the Children and Families Act 2014) [18]. The framework provides a foundation on which the core tenet of social work practice has developed, helping to illustrate the point within a ‘continuum of care’, in which experiences of care giving may become inappropriate [56] (p. 33) (Figure 3).

**Figure 3 healthcare-13-00280-f003:**
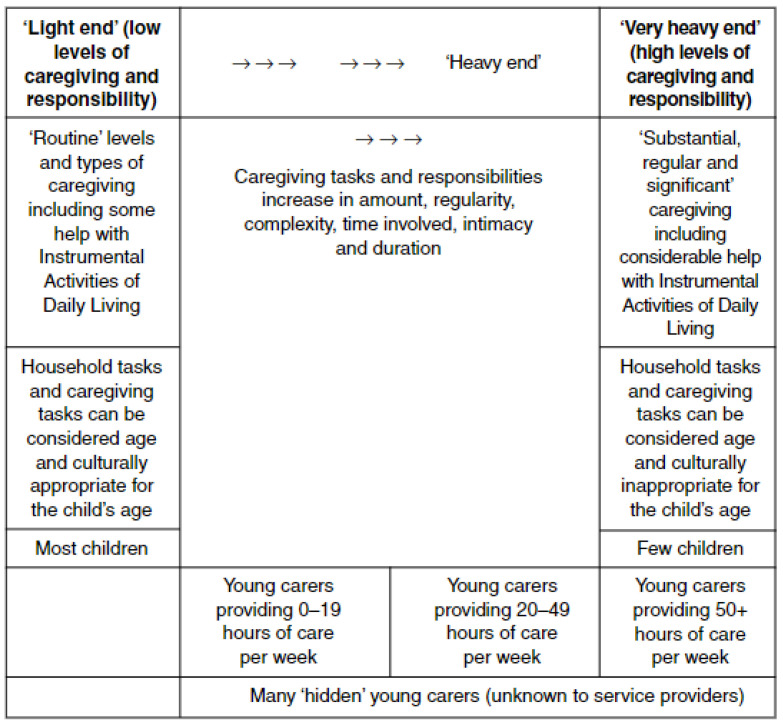
A continuum of children’s caregiving. Source: Becker [56].

Subsequent documentation throughout the UK reflects modern principles of safeguarding needs in guidance: *Working Together to Safeguard Children* [57], *Keeping children safe, helping families thrive* [58], *Working Together to Safeguard People* [59], *Co-operating to Safeguard Children and Young People in Northern Ireland* [60], *National Guidance for Child Protection in Scotland* [61] and *Getting it right for every child* (GIRFEC) [62].

These documents help to conceptualise risk within Jospeh et al.’s [10] (p. 84) conceptual concentric circle model of young caregiving. This illustrates the interplay of ‘caring for’, ‘caring about’ and the need of ‘care for’ young carers themselves. Its application is crucial in the development of preventative strategies offering whole family support and intervention.

In these contexts, understanding young carer’s early development may help to ‘untangle’ the dichotomous complexity of empowerment and risk associated with young caregiving [10] (p. 77). The multifaceted nature of the caregiving experience can in some cases foster resilience [63,64]. In contrast young carers face significant risk and emotional strain [65], poor physical and mental health [66,67,68] and compromised educational outcomes [8,37,69,70]. This complexity underscores the need for a nuanced understanding of individual circumstance, developmentally appropriate contexts, and recognition that in some cases empowerment and risk may coexist, influencing one another profoundly [71]. It is important to recognise that trauma itself is a socially rooted construct [72]. As such, professionals undertaking assessments of need must refrain from viewing the experience of YCEC through a subjectively informed lens of broader YC understanding.

Early childhood development lays the foundation on which all human experience is shaped [73]. A life course perspective is therefore imperative to inform multidisciplinary practice concerned with prevention of inappropriate caring responsibility. Such a perspective gives insight into mitigating the risks of associated biopsychosocial factors resulting from adverse childhood experiences (ACES) [74,75,76]. Danese and Lewis [77] (p. 108) describe early life stress as ‘hidden wounds’ of early childhood, contributing to poor health and mental health outcomes. Gabor Mate [78] further explains that ‘trauma is not what happens to you, it is what happens inside you’ [79] (p. 22). In this context, psychoneuroimmunology (PNI) (Figure 4) advances understanding of the relationship between the nervous system, psychological processes, and their effect on the immune system, strongly making a case for deeper and robust enquiry into the early life experiences of young carers.

Walker et al.’s [80] (p. 8) systematic review of young carer experience in Australia notably contextualizes the occurrence of being ‘born into caring’ as a distinct theme for analysis. Janes et al. [81] revisited past research in the context of psychosocial health. Their review concludes with the need for future research to engage with young carers of all age ranges, including those not already known to local authorities. Lacey et al. [66] strongly suggest the need for longitudinal studies which explore more deeply cause and effect within young caregiving relationship and wellbeing outcomes, further supporting the argument for greater focus on being and becoming a YCEC, and the wider biopsychosocial implications of such across the life course [82].

Prevalence: Growing prevalence of YCEC in the UK is evident, with reports illustrating increased awareness. For example, The Department of Education [83] reported an 80% increase in England in the identification of young carers aged between five and seven years between 2001 and 2011. Whilst incidence rates are thought to be underestimated, a further increase is evident between 2011 and 2021. Carers Trust [3] reported 3000 YCs aged between five and nine years spending at least fifty hours per week taking care of another person in England and Wales. In Northern Ireland, 1932 children aged between five and fourteen years were providing up to 19 h of care per week, with 316 providing more than 50 h of care per week [84]. Furthermore, census data for Scotland recognised children as young as three years old as having caring responsibilities [6], notably Scotland being the first nation in the UK to formally make provision to record children under the age of five years as caregivers within households in its census data.

Summary: Early childhood offers a critical window of development, affording a significant opportunity for early identification of risk to take place. Understanding concepts of caring and childhood in this context helps to address the ‘invisible vulnerabilities’ of being and becoming a YCEC [85]. The sharing of stories gives power to the young carer community, creating the potential for reciprocity [86]. Stories of experience help triangulate our understanding of prevalence data and quantitative research. Furthermore, understanding the perspectives of very young carers holds the potential to offer compelling insight to the broader field of mental health prevention, support, and intervention.

Aim: This scoping review is an initial step in quilting together a polyvocal investigation of YCEC [87]. The review is presented as part of a programme of PhD research undertaken by the lead author. It is intended to map the current knowledge base concerned with the lived experiences of the youngest carers in the UK. The objective is to create a step change in perceptions of young caregiving. This serves to better inform future developments of policy and practice in relation to possible mitigating factors concerned with early familial relationships and attachment.

With commitment to reducing research waste, the focus of this review centers on the qualitative representation of YCEC aged between birth and five years in the UK.

Review questions: How are Young Carers represented in the broader literature and what factors influence dominant representations of Young Carers in Early Childhood in the UK?

## 2. Design and Search

### 2.1. Design

This Scoping review has been undertaken in accordance with JBI guidance for scoping reviews [88]. A preliminary search of the JBI Systematic-Review-Register [89] was conducted and no current or underway systematic reviews or scoping reviews of the specific topic were identified.

An a priori protocol was developed and published (Systematic-Review-Register—Systematic Review Register|Joanna Briggs Institute (Adelaide, South Australia), 29 February 2024) entitled *Young Carers in Early Childhood—Scoping Review*. The lead author adhered to the PRISMA-ScR checklist for scoping reviews (Figure 5) [90,91]. The PCC Participant, Concept, Context approach defined the eligibility criteria in relation to participants. This defined the age of young carers included in studies (0–5 years), the definition of a young carer and the location of young carers. In this case, included qualitative studies were conducted with participants from the UK only.

### 2.2. Search

The search strategy was devised in consultation with knowledge users and a library information specialist to define clear, succinct search terms. The search took place between 13–18 September 2024. Examples of key words and phrases include ‘Young carers’, ‘Early Childhood’ and ‘Lived experience’. Key words and controlled vocabulary were utilised to manually adapt the search strategy for online databases. These included publications dated 2014–2024 in Scopus, PubMed, PsycINFO, Eric, Web of Science, and the Carers Trust and NSPCC websites. Manual searches of reference lists, including past systematic reviews concerned with young carers, were undertaken. Articles were imported to Rayyan [91]. Duplicates were removed before screening.

An independent reviewer screened 10% of all articles, including title, abstract and participant age range where indicated in the methodologies. Full text screening proceeded, with a further 10% screened by the same independent reviewer. Any conflict was arbitrated by the lead author in consultation with knowledge user representatives, by referral to the inclusion and exclusion criteria. Results were discussed with knowledge users at all stages of the review process, including defining the aims, sharing articles for consideration and conclusions [88]. Eligible sources for inclusion included peer reviewed primary research articles and internet searches of grey literature. Exclusion criteria included articles where samples were indivisible by age (e.g., “Participants ranged in age from 5 to 12 years”) or the age range of participants were not specified. Opinion pieces and local service data were excluded. Relevant systematic reviews were searched for any additional eligible articles within reference lists not captured by the original search.

## 3. Results

A search of Scopus, PubMed, PsycINFO, Eric, Web of Science databases, and the Carers Trust, NSPCC, yielded 865 articles, imported to Rayyan [91]. After removal of duplicates, 669 remained and were subject to title, abstract and population screening, 10% of which were blind screened by an independent reviewer. After excluding irrelevant articles, 17 full text articles were retrieved for full text screening, of which 10% were screened by the independent reviewer. Of these, eight serious case reviews (SCR’s) were screened from the NSPCC database. The results were discussed with knowledge users to reach a final agreement.

## 4. Discussion

A comprehensive search of the literature yielded no studies which met the inclusion criteria for this scoping review, the exception being a yet unpublished master’s degree dissertation [92], entitled *Capturing the ontology of young carers in the early years—Threads of hope through lived experience*. This work has not been included due to the ongoing nature of the research. The work explored the lived experience of two families, each centering on a young carer aged four years old. As the work has yet to be subject to peer review, it has been excluded in this instance. Other studies were excluded due to the specified age range criteria and the geographical area of the research participants. The absence of relevant research highlights a significant gap in knowledge regarding the way in which the lived experiences of young carers in early childhood are represented and understood.

Themes observed during the screening process are discussed within this narrative analysis of excluded articles and broader representations of young carers. Concurring with Janes et al. [81], existing empirical research presents subjective outputs. Participants within the research are mostly already known to services. As such, participant age range is dominated by experience of children grouped within age ranges of 5–18 years., with greatest representation from those aged 7–18 years (in the context of childhood) [93]. As a result, outputs lack the voice, developmental and pedagogical perspective of YCEC.

This highlights a compelling need for further research specifically concerned with YCEC, originating from all disciplines. Furthermore, research-informed guidance is required to develop policy and practice to further represent the voice of YCEC. This is particularly needed in the context of whole family support and for local authorities wishing to reach and engage with young carers who may be viewed as more challenging to serve.

Advice and guidance: Limited research and guidance was found to be available to support service provision in assessing and meeting the needs and transitions of YCEC. Whilst independent documentation is often available within local services, only one overarching document was found, consistently recommended to the lead author by knowledge users [94]. The guidance was excluded from the review due to age range exclusion and revisions made by the regulatory body in England since publication [95]. A resource pack for schools aimed at the Early Years Foundation Stage (EYFS) and key stage 1, produced by Caring Together [96], provides some practical support and activity ideas for schools, but refrains from voicing the experience of the, YCEC in broad developmental terms, fully relating to the EYFS (0–5 years.) [97].

National organisations offering effective awareness training to education providers throughout the UK do not yet extend that guidance specifically to the ECCE sector (for example, Carers Trust and The Children’s Society [98]. Provision of targeted Young Carer support is itself variable across the UK. For example, some local authorities endorse a whole age model of delivery, whilst other parts of the UK offer tailored support to young carers from predetermined age ranges [7,8].

Where policies and support systems fail to fully capture the full age range of young carers, a distorted representation influences prevalent narratives. Phelps [5] noted the importance of giving voice to society’s most vulnerable young carers, noting that their stories should not be obscured by those more commonly represented. As a result, information sharing between the ECCE sector and primary schools regarding vulnerabilities is not well informed, nor is a common language for ‘Young Carers’ used to better recognise YCEC upon transition into primary school. Consequently, opportunities are sometimes missed by professionals to identify at the earliest opportunity very young children who are, or are likely to become, young carers. In addition, there is little evidence of professional engagement across disciplines to consider the barriers to identification and support for YCEC.

Defining the age of assessment and support: Gibson et al. [99] acknowledge young carers from birth and highlight that the undertaking of assessment may come from all the professionals concerned with a family, including an Early Help Assessment in England, correctly observing that there is no lower age limit for an assessment of need in the context of young carers. However, it was noticed during screening exercises that legislative frameworks are at times misrepresented in the literature. For example, Darling et al. [100] (p. 532) and Waters [101] (p. 1) insert age limitations when reporting legislation pertaining to the protection of young carers’ rights, incorrectly creating exclusion criteria for the early identification of YCEC. As such, decision makers are not as well informed in recognising YCEC as they could be, feeding narratives of implicit bias towards families in which young children provide care to another person. Such misrepresentation is unhelpful when setting commissioning priorities, contributing to a postcode lottery of support across the UK [13]. Furthermore, discussion regarding the purpose of early help, and family support strategy more broadly, highlights the need for greater service evaluation, particularly concerning the modus operandi of young carers’ support services in the broader sphere of family support, which most commonly seek to intervene when problems already exist, rather than preventing crisis from occurring [102].

Language and terminology: In 2016, a report from the Office of the Children’s Commissioner for England made a commitment to better understand the experience of society’s youngest carers [103]. The report prompted better advocacy for greater focus on identification and support for YCEC, up to and including children aged 8 years. Pathways to identification and assessment of YCEC were viewed through a lens of safeguarding, with the report suggesting that services were not able to support the needs of young children, due to lack of investment. Caregiving in this context was defined into three categories: ‘minimal’, ‘low-level’ or ‘excessive’. As such, YCEC were considered to be children in need [55] or placed in the care of social services. Local tabloids sensationalised the report, with headlines of ‘stolen childhood’ labelling YCEC as infant carers, feeding narratives based on adult-centric views of young caregiving [104]. Infancy in the developmental context of early childhood is defined as the period from gestation up to 2 years. [27]. As such, barriers may have prevented societal acceptance of YCEC as providers of care. Instead, greater collaboration of cross-policy initiatives could be developed to improve support for families during infancy, including pre-birth assessment and post birth support throughout early childhood. Current research offered by Burch et al. [105] (p. 23) has found that, in two hundred recent care proceedings concerned with infants under twelve months, 34% involved a parent with ‘learning disabilities or learning difficulties’. Cooccurring risk factors included substance abuse, mental health, and domestic abuse, raised by professionals. Long-term, whole family support in this context is necessary to mitigate the risk of children being taken into care, and to provide more targeted support for those families where the care of another person is most likely to be provided, in some way, by the child, as such, making the case to extend professional awareness training concerned with young carers to all professionals responsible for the health, care and education of children and their families in a child’s early years.

There are long standing arguments, of course, which view definitive terminology as problematising childhood [106,107]. This argument, in the context of young carers, is countered by Joseph et al.’s [12] (p. 44) concentric circle model, helping to address the nuances of young caregiving. The model offers opportunity to reframe resistance to labelling YCEC or viewing young carers as a homogenous group. Saragosa et al. [1] suggest that broader age group representation in the context of young adult carers is a result of sociodemographic changes. Further research is needed to understand such changes in relation to caregiving identified in early childhood. Additionally, research is required to understand society’s perception of young carers in the context of early childhood.

Safeguarding: In recent years, serious case reviews (now called Child Safeguarding Practice Reviews (CSPRs) in England) concerned with young carers have highlighted notable factors requiring further exploration in the context of YCEC. The NSPCC [108] published a report highlighting a lack of safeguarding training amongst early years staff, and lack of robust policies and procedures with which staff must comply and must implement in practice. Where concerns about children were raised, these were not consistently, nor appropriately, shared. Of greatest pertinence to YCEC if that, where support needs of families were recognised, staff failed to fully acknowledge the impact of such on parenting capacity. Too often, ECCE settings do not receive vital information from other services involved in family support and, as a result, professionals are not well informed of a child’s unique history and wider social environment. Consequently, early years practitioners are not always sufficiently well informed to protect children or to prevent risks that were otherwise known. The report makes clear recommendations which advocate for ‘professional curiosity’ in all cases, imploring the sector to ‘build up a picture of a child’s lived experience and show curiosity about their life outside of the setting, such as their home environment and family relationships’, NSPCC [108] (p. 4). Poor understanding of lived experiences in early childhood in the context of young caregiving creates barriers to having relevant experiences and needs heard and understood. As a result, insufficient recognition of family history inhibits assessments, which must consider the cumulative impact of factors associated with young caregiving. Where needs are poorly understood, application of thresholds within Early Help, child in need assessments and family support strategies are brought into question [109], particularly when young carers are exposed to multiple risk factors concerned with caregiving, including a trigger trio of risk, defined as parental mental health, addiction, and exposure to domestic violence [110].

A preliminary search of serious case reviews (SCRs) available on the NSPCC database provided explicit examples of too little, too late. Young carers were identified as an outcome of a SCR, but not always by those professionals involved in the care and support of the child/ren and their families leading up to the review. This highlights a significant lack of awareness and understanding of young carers in cases of children in need. For context, one in every thirty children in England are reported to be children in need [111]. The following graphic (Figure 6) helps to illustrate the current percentage of children on a child in need plan in England, and the factors of concern regarding a parent or other person. The percentage of children identified as young carers is marginal; furthermore, this figure has decreased by −0.80% since 2023, despite incidences of child in need plans increasing overall.

Furthermore, 22% of all children in Northern Ireland face severe hardship [112], 20% of children receiving care and support in Wales are under 5 years old [113] and 24% of all children in Scotland are living in poverty [114]. These multiple hardships faced by children in the UK are further exemplified when caring responsibility is left undetected. Furthermore, the Safeguarding Practice Review Panel [115] (p. 74) suggests that the lack of whole family approaches to assessment of need can lead to subjective views towards individuals and risk. Subsequently, when vulnerabilities are viewed in isolation, the person of focus is often seen as ‘the risk’ rather than viewing that person and all family members concerned as ‘at risk’. As such, current methods of assessment are, at times, failing to protect young carers and the rights afforded to them. Little consideration is given to the powers that operate in a family’s life and the subsequent threat presented by such [116]. Future assessments should provide ‘broad and holistic’ considerations of need which are meaningfully centered on the ‘voices and lived experience of all relevant children (those subject to support and protection as well as other related children in the family)’ [115] (p. 78).

Other risk factors concerned with outcomes from SCR/CSPRs involving young carers, noted exposure to suicide, exposure to bereavement, failings of professionals to acknowledge a child’s family history, and a culture of failing to adhere to evidenced based practice, (for this reason local practice guides have been excluded from this scoping review in the context of YCEC). Where children and young people were recognised as young carers, mental health and primary health care services failed to recognise the significance of their vulnerability. Tragically resulting in cases of young carers dying by suicide, and/or neglect, or being placed into the care of the local authority. A child’s voice should unarguably be central to all assessment and information concerning the needs of a family. Evidence strongly indicates this is not consistently the current case for YCEC.

In summary, research screened as part of this process has been largely led by experts outside the field of ECCE. Key empirical research regarding the assessment of young carers has yet to include the informed developmental perspective of YCEC in successful models [117]. As a result, framing of experience has led to a technocratic approach to young carers’ policy which may benefit further from crucial pedagogical exploration [118]. ECCE is one of the UK’s vital public services, regarded as a foundation within society, alongside education, social care, and health [119]. A fundamental bedrock of whole family support deserves better inclusion in broader narratives of young carers’ care and education. In doing so, the UK will be initiative-taking in leaning further towards a goal of achieving improved levels of response for all young carers, continuing to explore the voice of childhood in its entirety. This will help to broaden “extensive awareness at all levels of government and society of the experiences and needs of young carers” [10] (p. 81) and assist in ensuring that future assessments of need reach those who are currently left behind [32].

### 4.1. Implications for Research

To address knowledge gaps identified in this review, recommendations for further research concerning the voice and experience of YCEC are made. Research concerned with young carer visibility within society should be constructed in the context of the spaces and places encountered by YCEC, including the early childhood care and education sector. This would support the development of whole systems approaches which foster prevention and, where appropriate, sustained support throughout the life course [120], consequently aiding earlier awareness of young carers, broadening young carer narratives, and furthering policy and practice developments in the UK to support systems change in the future [16].

### 4.2. Limitations

This scoping review is positioned within the context of UK society, legislation, and policy. As a result, the review presents a Eurocentric position. Consequently, the review may not be generalisable in broader international comparison [121].

The concept of parentification was understood as a known risk and consequence of young caregiving, therefore the subject was not explored explicitly within this review, but recognised in the context of perceived inappropriate caring and protecting a child’s right to care. For further information, Dariotis et al. [122] have undertaken a comprehensive systematic literature review of the subject, Hendricks et al. [123] conducted a parentification concept analysis amongst young carers, and Sharpe [124], whose thesis explored *Parentification: Identifying Young Caregivers at Risk* is recommended. 

## 5. Conclusions

This scoping review sought to assess the extent of literature specifically concerned with the lived experiences of YCEC. The authors highlight a gap in research concerned with the voice and representation of YCEC in the UK. Whilst challenges are presented, understanding early childhood development, and particularly early attachments and companionship in all young carer’s lives, could help better inform assessments of need where pathologies of trauma exist.

By defining the concept of YCEC, its distinct developmental importance in the context of young caregiving is represented. The significance of this highlight’s dominant representation of young carers in the UK, shaped by an amalgamation of economic, social, cultural, and institutional factors.

The legislative rights afforded to young carers within the UK apply to children and young people from birth to eighteen years (in the context of childhood). There is no lower age limit placed upon an assessment of need in any UK legislation. Strategies, which inform practice to help young carers self-identify and enhance professional awareness to identify young carers at the earliest opportunity, can be further improved. Providing research that plays a role in equipping all sectors with a deep understanding of the ontology of YCEC is therefore instrumental in shaping economic growth and better outcomes for society.

## 6. Knowledge User Comment

‘This review clearly demonstrates the urgent need for a greater focus on the youngest young carers. We know from the 2021 Census that over 3000 children aged just 5 to 8 years old were recorded as caring for more than 50 h each week, in England and Wales yet we know very little about their distinct experiences or support needs, including those in their early years.

If we are to ensure young carers are identified at the earliest opportunity, then there is a real need to focus on how we better identify and support young children who have caring responsibilities. This requires a focus across research, policy, and practice’. (Andy McGowan, Policy and Practice Manager, Carers Trust).

## Figures and Tables

**Figure 1 healthcare-13-00280-f001:**
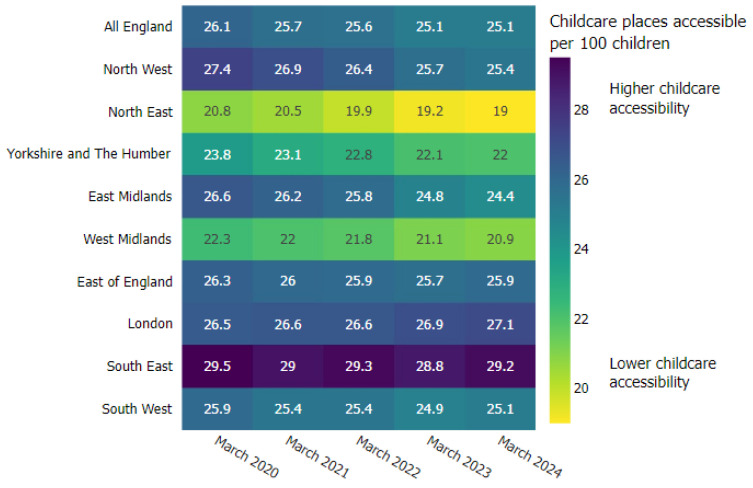
The change in childcare accessibility by region over time, England. Source: Commentary: Changes in access to childcare in England—https://www.gov.uk/government/publications/changes-to-access-to-childcare-in-england/commentary-changes-in-access-to-childcare-in-england (accessed on 18 October 2024) [34].

**Figure 2 healthcare-13-00280-f002:**
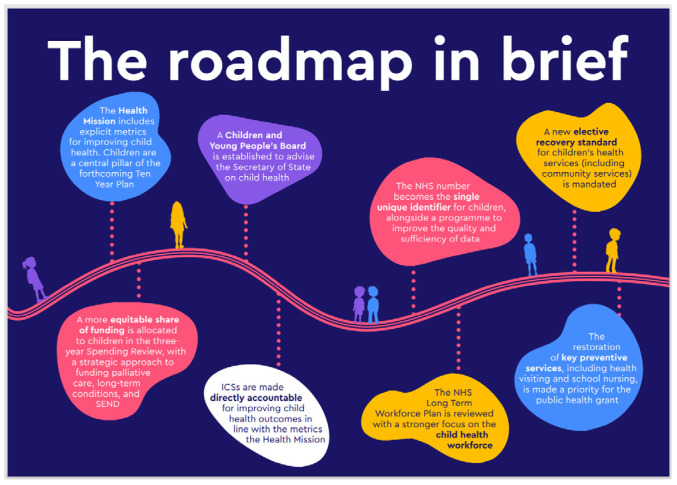
The healthiest generation of children ever: A roadmap for the health system. Source: Children and Young People’s Health Policy Influencing Group [36].

**Figure 4 healthcare-13-00280-f004:**
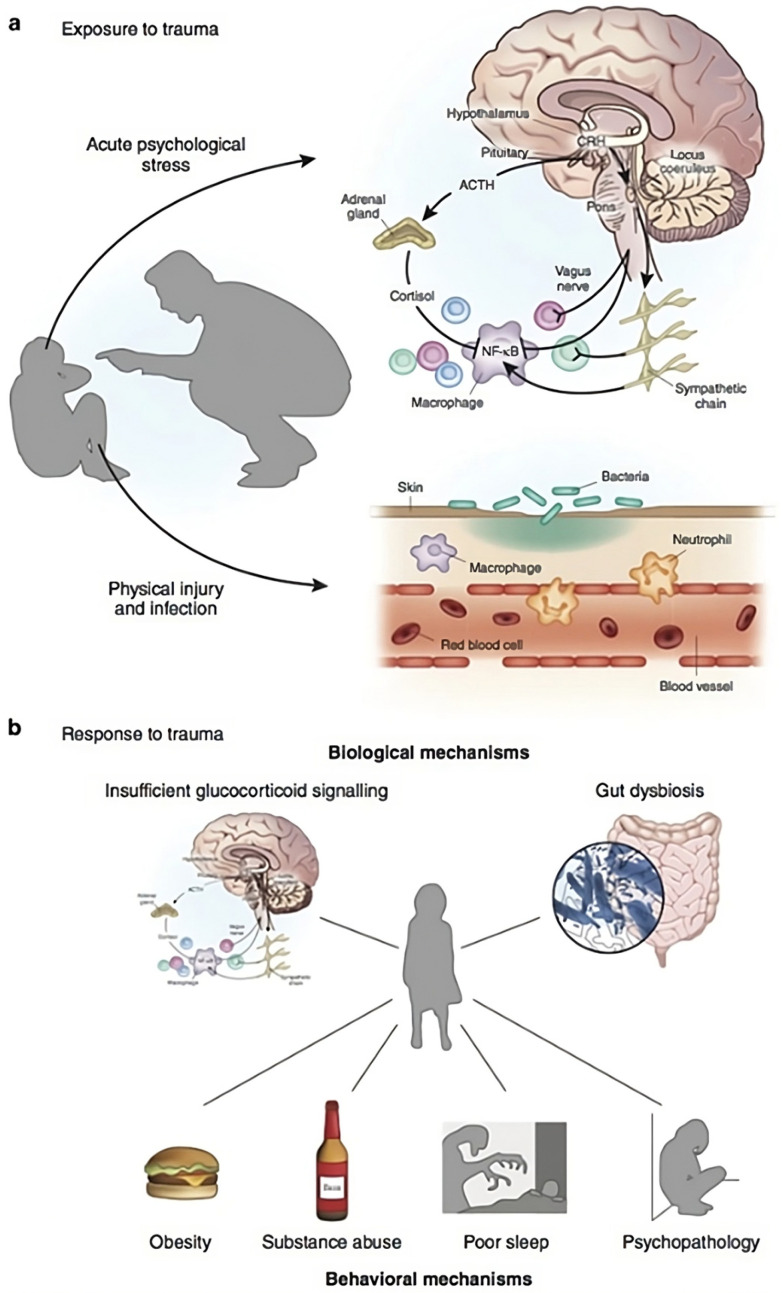
Psychoneuroimmunology of early Life Stress: Exposure to trauma. Source: Danese and J Lewis [77] (p. 106). This attribution should clearly indicate that this content is licensed by Springer Nature Customer Service Center GmbH, and is NOT part of the overriding OA/Creative Commons license.

**Figure 5 healthcare-13-00280-f005:**
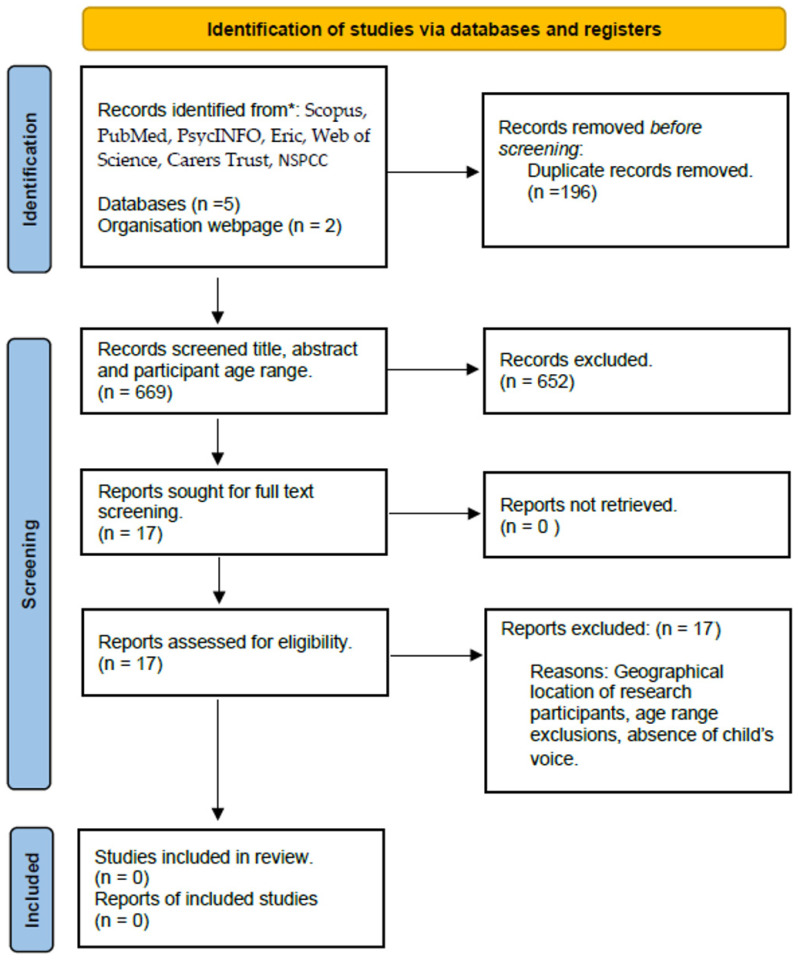
PRISMA chart, Scoping Review. ‘n’ denotes number of studies or records at each phase of review process. * denotes databases used.

**Figure 6 healthcare-13-00280-f006:**
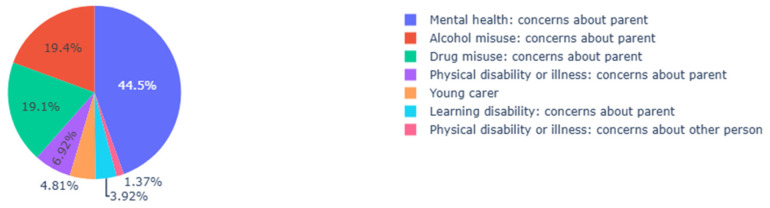
Department for Education Child in Need Assessment Data. (England). Source: Statistical data Department for Education, [58].

## Data Availability

All data used in this scoping review are derived from publicly available sources as referenced within the article. No new data were created or analyzed.

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
