# Peer review of "Young Carers in Early Childhood—How Are Young Carers Represented in Broader Literature and What Factors Influence Dominant Representations of Young Carers in Early Childhood in the UK?"

_healthcare, 2025, doi:10.3390/healthcare13030280_

Round 1
Reviewer 1 Report
Comments and Suggestions for Authors
Thank you for the opportunity to review this paper of great importance. This issue is silent in much of the social services narrative, and you are bringing this to light with a foundational scoping review and a pathway for research ahead.
I suggest minor changes only.
1. The introduction was slightly unclear in with the point on access to ECE. I believe the authors are making a connection between access to ECE and a fairer life for children who are carers by bringing them into a system but this was not entirely clear.
2. More clearly define, in the introduction, what is meant by Carers and the identification of tasks associated with caring at a very young age. This has been done in Figure 5 and a definition comes on P7, but needs to be clear from the start what is meant by young carers.
3. P 2 Line 84 "childcare desserts" should be deserts.
4. Suggest use of less figures as they are at times distracting-be clear on reasons for inclusion.
5. Figures 2 and 3 are more difficult to read but suggest enhancing the readability of all figures.
Author Response
Thank you for your feedback, it has been gratefully recieved and considered.
- The introduction was slightly unclear in with the point on access to ECE. I believe the authors are making a connection between access to ECE and a fairer life for children who are carers by bringing them into a system but this was not entirely clear.
Thank you. I have expanded the sentence ( Line 132) to help clarify the point.
- More clearly define, in the introduction, what is meant by Carers and the identification of tasks associated with caring at a very young age. This has been done in Figure 5 and a definition comes on P7, but needs to be clear from the start what is meant by young carers.
Thank you. I have moved the 'definition in context' section forward to support the context and flow of the introduction ( Line 87).
- P 2 Line 84 "childcare desserts" should be deserts.
Thank you !! This has been corrected.
- Suggest use of less figures as they are at times distracting-be clear on reasons for inclusion.
- Figures 2 and 3 are more difficult to read but suggest enhancing the readability of all figures.
Thank you for this observation. I have made alterations to the readability.
Thank you once again.
Kind regards
Carly Ellicott
Reviewer 2 Report
Comments and Suggestions for Authors
I enjoyed reading this paper and ask that it be accepted.
Abstract is very detailed and gives a glimpse of the entire study that discusses the background, aim, method and results of the study.
Young Carers is an important topic but not studied a lot in the literature. So, undertaking a systematic review on this topic is important. And that is exactly what the researchers have done.
At the outset the understanding of early childhood years and what it means in UK is well explained. The visual models within the paper aids further understanding of this complex topic. Thus, the literature review is well laid out.
Researchers clearly define the term “Young Carers” and then explain why this topic becomes even more important at the early childhood level. The rationale provided for the same is very sound.
Thus, the overarching research question makes complete sense: Review questions: How are Young Carers represented in broader literature and 276 what factors influence dominant representations of Young Carers in Early Childhood in 277 the UK?
The search strategy utilized to go through and capture all different articles related to this topic, is very sound. I enjoyed seeing a clear partnership between the researchers and library specialists.
Although, they couldn’t find a lot on this topic, due to the topic of choice. They stated in their conclusion that it is imperative to study this topic further, which I completely agree with. Implications for further researcher and limitations are well described.
Overall, I see a lot of merit in this paper.
Author Response
Thank you for your positive and constructive feedback, it has been greatly appreciated.
Kind regards
Carly Ellicott
Reviewer 3 Report
Comments and Suggestions for Authors
Dear Respectable Authors
Thank you for considering this great area of research related to Young Carers in Early Childhood. You conducted a scoping review to map the current qualitative knowledge base concerning the lived experiences of the youngest carers in the UK. Your results are interesting but the way you report the manuscript needs some revisions as follows;
- Abstract, please remove all subheadings from the abstract section.
- Abstract, lines 24 and 33, please remove the reference from the abstract.
- Abstract, please add the exact time of the search or the exact period of the search not only the year but also the date.
- Abstract, please add abrif eligibility criteria to this section.
- Main text, please check the reference style with the author guidelines.
- Introduction, Figures 1-6 are not clear enough. Please add more transparent figures.
- Methods, please revise your methods section. You missed the essential steps of a scoping review in your work. Please take a look at PRISMA-ScR and modify your text based on it. Add information regarding study selection, eligibility criteria, data charting process data item, and methods of synthesis of the results.
- Discussion section, you need to have a discussion section in your manuscript and discuss your results with others in this field. Some of these discussions have been made in the results section, but they need to be separated.
Cheers
Author Response
Thank you for your constructive feedback, i have made the following changes and considerations:
1.- Abstract, please remove all subheadings from the abstract section.
- Abstract, lines 24 and 33, please remove the reference from the abstract.
- Abstract, please add the exact time of the search or the exact period of the search not only the year but also the date.
- Abstract, please add a brief eligibility criteria to this section.
The abstract has been reduced in line with the recommended word count. Please see the revised submission.
2.- Main text, please check the reference style with the author guidelines.
I have submitted the review in line with the open format recommended guidelines.
3,- Methods, please revise your methods section. You missed the essential steps of a scoping review in your work. Please take a look at PRISMA-ScR and modify your text based on it. Add information regarding study selection, eligibility criteria, data charting process data item, and methods of synthesis of the results.
Please see the attached document.
: Preferred Reporting Items for Systematic reviews and Meta-Analyses extension for Scoping Reviews (PRISMA-ScR) Checklist
4. - Discussion section, you need to have a discussion section in your manuscript and discuss your results with others in this field. Some of these discussions have been made in the results section, but they need to be separated.
Please see the revised submission, I have altered the results section to move all discussion clearly into the 'Discussion' section.
Kind regards
Carly Ellicott

Round 2
Reviewer 3 Report
Comments and Suggestions for Authors
Dear Respectable Authors
Thank you for your clarifications.
Cheers